# Labeled TEMPO-Oxidized Mannan Differentiates Binding Profiles within the Collectin Families

**DOI:** 10.3390/ijms232416067

**Published:** 2022-12-16

**Authors:** Florent Le Guern, Anne Gaucher, Gina Cosentino, Marion Lagune, Henk P. Haagsman, Anne-Laure Roux, Damien Prim, Martin Rottman

**Affiliations:** 1Institut Lavoisier de Versailles, CNRS, UVSQ, Université Paris-Saclay, 78035 Versailles, France; 2Faculté de Médecine Simone Veil, Université de Versailles St Quentin, INSERM UMR U1173, 2 Avenue de la Source de la Bièvre, 78180 Montigny le Bretonneux, France; 3Section Molecular Host Defence, Division Infectious Diseases & Immunology, Department of Biomolecular Health Sciences, Faculty of Veterinary Medicine, Utrecht University, 3584 CS Utrecht, The Netherlands; 4Hôpital Raymond Poincaré, AP-HP, GHU Paris Saclay, 104 Bd Poincaré, 92380 Garches, France; 5Plateforme des Biomarqueurs Innovants, 104 Bd Poincaré, 92380 Garches, France

**Keywords:** mannan, oxidation, fluorescent, avidity, binding, lectin, tempo, surfactant protein D

## Abstract

Establishing the rapid and accurate diagnosis of sepsis is a key component to the improvement of clinical outcomes. The ability of analytical platforms to rapidly detect pathogen-associated molecular patterns (PAMP) in blood could provide a powerful host-independent biomarker of sepsis. A novel concept was investigated based on the idea that a pre-bound and fluorescent ligand could be released from lectins in contact with high-affinity ligands (such as PAMPs). To create fluorescent ligands with precise avidity, the kinetically followed TEMPO oxidation of yeast mannan and carbodiimide coupling were used. The chemical modifications led to decreases in avidity between mannan and human collectins, such as the mannan-binding lectin (MBL) and human surfactant protein D (SP-D), but not in porcine SP-D. Despite this effect, these fluorescent derivatives were captured by human lectins using highly concentrated solutions. The resulting fluorescent beads were exposed to different solutions, and the results showed that displacements occur in contact with higher affinity ligands, proving that two-stage competition processes can occur in collectin carbohydrate recognition mechanisms. Moreover, the fluorescence loss depends on the discrepancy between the respective avidities of the recognized ligand and the fluorescent mannan. Chemically modulated fluorescent ligands associated with a diversity of collectins may lead to the creation of diagnostic tools suitable for multiplex array assays and the identification of high-avidity ligands.

## 1. Introduction

Sepsis is characterized by a dysregulated response of the host to infection, which causes a pathological process leading to end-organ dysfunction, shock, and ultimately death. It is a major healthcare concern and is a major driver of infectious disease mortality in developed and less developed countries, and the World Health Organization has made sepsis a global priority. While the recent SARS-CoV-2 pandemic has shown that viral infections can be a cause of sepsis, most of the etiologies of sepsis are of bacterial or fungal origin. Most early sepsis diagnosis tools identify elements of the host response and can be more or less specific. However, these parameters are not diagnostic in regard to the bacterial or fungal infection itself. Moreover, given the worldwide emergence of antibiotic resistance, the reasoned usage of antibiotics is cautioned, and while the earliest adapted treatment is mandatory in sepsis patients, distinguishing between patients that do not have an infection and should not receive antibiotics and those who have an infection and should receive antibiotics is becoming more and more important in the current context [1]. Regarding immunity, the innate immune system is tasked with the destruction of pathogens based on the recognition of pathogen-associated molecular patterns (PAMP), which are molecular motives typical of pathogens and are not found in the host, thus sparing beneficial microbiomes. Studies of the innate immune system show that lectins play a major role in pathogen recognition [2,3]. These proteins have several regulatory functions in biological systems, such as cell adhesion [4], protein controls [5], and immune protection [6]. Indeed, these proteins are able to recognize distinct carbohydrate patterns, which act as a cellular identity for pathogenic bacteria. Therefore, some are present as soluble proteins or on the surface of immune cells, such as dendritic cells and macrophages. Consistent with the wide carbohydrate diversity, several soluble lectin families have been described, such as the well-studied family of collectins that includes the mannan-binding lectin (MBL) and surfactant proteins A or D (SP-A or D). Their capacity to recognize microbial glycans is ensured by a specific domain, known as the carbohydrate recognition domain (CRD), connected to a collagen-like stem by a flexible neck domain. The operational principle of CRD hinges on the complexation between amino acids, glycans, and cations, especially calcium ions. Most lectins exist as multimers, which increase their avidities for the bacterial pattern through multivalent bindings [7]. Whereas the most common structure of MBLs is composed of three to five trimers (nine to fifteen CRDs) in the form of a bouquet, SP-Ds have a cruciform shape associating four trimers (twelve CRDs) (Figure 1) [2]. Collectins are then able to recognize notorious opportunist bacterial pathogens, such as *Escherichia coli*, *Staphylococcus aureus*, and *Streptococcus* strains [8].

The immune activity of collectins is based on CRDs, which present high selectivity for diequatorial vicinal hydroxyl groups through calcium cation complexation [9,10]. These stereochemical specificities can be found within the structures of carbohydrates, such as D-glucose, L-fucose, and D-mannose (Figure 1). In fact, the structural oligomerization of lectins also influences their binding avidities and specificities. Then, in order to characterize the carbohydrate-binding profile of each lectin, avidity screenings are achieved using glycan arrays. Overall, collectins present a high avidity for D-mannose and especially for L-glycero-D-manno-heptose (Heptose, Figure 1). The high level of interaction of this latter carbohydrate is due to the participation of a supplementary vicinal hydroxyl couple at positions C6 and C7 (Figure 1) [11]. Thus, various pathogens, such as Gram-negative bacteria, are recognized by CRDs because of heptose moieties within the lipopolysaccharide (LPS) structures [11,12]. Polymannose (**Mannan**), and especially those with α(1-2)- and α(1-6)-linked mannose units (from yeast, for example), are obviously recognized by CRDs [13]. In fact, mannan derivatives were even developed in order to image immune cells [14,15]. Recombinant lectins have also been used for their ability to capture specific glycan patterns along biochemical applications, such as chromatography [16], blood treatment [17], or immune assays [18]. Fluorescence polarization-based competitive binding assays were developed in order to evaluate the actual capacity of inhibitors of bacterial lectins, which are involved in biofilm formations [19,20]. Cartwright et al. proposed an enzyme-linked lectin sorbent assay (ELLecSA) to detect PAMPs in whole blood [18]. By introducing engineered and supported MBLs into blood samples, endotoxins have been caught, extracted, and revealed using an ELISA-like process. Thanks to the use of an optimized MBL, they were able to detect PAMPs at a concentration of 15 ng/mL in samples. The concentration in blood in clinical situations is around 500 pg/mL [21]. The detection of PAMPs in the bloodstream during systemic infections is an intriguing concept warranting additional investigations [22].

The original purpose of this study was to create a simple support that may lose its fluorescence when in contact with endotoxins and may be adapted to multiplex bead array assays. Recently, these assays showed their utility by offering rapid, precise, and cost-effective quantifications of soluble analytes from serum and plasma samples [23]. These analyses are based on fluorescence variations following the contact between beads and biological fluids. To reach the objective, we suspected that a novel two-stage competition assay could be designed by using a chemically optimized fluorescent ligand. We thus proposed the preparation of novel fluorescent derivatives with defined avidities, which were to be initially caught by lectins and then displaced by better-recognized ligands. These derivatives had to fit with the following specificities: (1) their avidities for collectins had to be weaker than those of PAMPs, and (2) their avidities needed to be higher than those of simple glycosides to ensure their stabilities in clean biological samples. Therefore, their preparation had to include a chemical tool able to modulate the avidity. We decided to use oxidized mannan (**Mox**) obtained by processing TEMPO oxidation. While oxidized mannan is already used for the preparation of fluorescent derivatives, there is a lack of data describing the actual involvement of the mannuronic acid moiety in the SP-D- and MBL-recognizing mechanism. A new tailor-made fluorescein was chemically bound to **Mox** to create fluorescent mannan (**Mf**). After characterizing the avidities of all compounds, the system was challenged through competition assays to ascertain whether a two-stage competition occurred and whether the system could actually be used to detect PAMPs in biological samples (Figure 2).

## 2. Results

### 2.1. Preparation and Characterization of TEMPO-Oxidized Mannan

To create the fluorescent ligand, mannan from *Saccharomyces cerevisiae* was first oxidized in order to implant carboxylic acid functions in the chemical structure for further couplings. A protocol based on the TEMPO-catalyzed oxidation was adapted (Figure 3).

Commercial yeast mannan was then dissolved in an aqueous mixture of TEMPO, sodium bromide, and hypochlorite sodium. Both temperature (3 °C) and pH (9.3) must be strictly controlled in order to prevent any degradation of the mannan backbone [24]. After the iterative additions of sodium hydroxide over 2 h, no change in the pH was observed. Using a borohydride sodium mixture as the reducing agent, the reaction and excess oxidative reactants were quenched. After the purification steps, the freeze-dried and targeted oxidized mannan was obtained as a white powder without a significant loss of mass (90% of initial mass). Both IR and ^13^C NMR analyses undoubtedly confirmed the oxidation of mannan, displaying characteristic stretching bands at 1600 cm^−1^ (Appendix A) and the signal at 176 ppm (Appendix A), respectively, in accordance with the presence of a carboxylic acid fragment. Once the oxidation was confirmed, a high-performance liquid and size exclusion chromatography with multi-angle light scattering (HPLC-SEC-MALS) analyses was achieved in order to estimate the polymeric degradation (Appendix A). Even if the chromatographic profiles of **Mannan** and **Mox** are different because of aggregations [14], MALS allowed the estimations of molar mass repartition at around 26 and 28 kDa, respectively. Overlapping makes the clear characterization of the exact molar mass repartition difficult, but this result clearly indicates no substantial mass reduction, and thus, no significant degradation of mannan occurred during the TEMPO-catalyzed oxidation.

In order to determine the evolution of the degree of oxidation (DO) during the oxidation course, a kinetic study was carried out (Figure 4). To perform this study, carboxylic acid contents in **Mox** were estimated using conductimetric titration according to previous research [25]. During the reaction, samples were extracted, quenched using reducing agents, and finally titrated. It appeared that the maximum DO was achieved after two hours and was about 25%. The rate, following first-order kinetics, is about 4.2 × 10^−4^ s^−1^, and the maximal carboxylic acid concentration attained was 1.4 mmol/g. As a result, mannan oxidation can be controlled, and **Mox** with a specific DO can be easily prepared by quenching the process at the right moment.

### 2.2. Preparation of Fluorescent Mannan

A novel fluorescein derivative was designed and synthesized to create the fluorescent ligand (**Mf**) (Figure 5). The purpose of this synthesis was to include a primary amine function within the chemical structure of the fluorophore and then to couple it with the carboxylic acid functions of **Mox**. We proposed the direct inclusion of a spacer inside the methyl ester fluorescein structure (**2**) using a Williamson coupling. The esterification, followed by Williamson-type etherification, was realized, leading to NH(Boc) derivative **3** in 70% yield (three steps). Finally, Boc cleavage under acidic conditions led to quantitative ammonium chloride **4**, which was characterized by NMR and MS analyses (Appendix A). Photophysical analyses demonstrated that the chemical transformation led to a serious loss of fluorescence (Φ_f_ = 0.32 in NaOH 0.1M) but was still efficient enough for a fluorescent probe (Appendix A). Finally, the **Mf** was prepared by using the 1-ethyl-3-(3-diméthylaminopropyl)carbodiimide (EDC)/N-Hydroxysuccinimide (NHS) protocol and 1.5 eq of **4**, whereas 1 eq was directly related to the titrated carboxylic acid content in **Mox**. After purification and freeze-drying, the **Mf** was obtained with 50% mass conservation, and absorbance properties estimated a degree of substitution average of about 10%, following the Beer–Lambert law. Despite the low yield of the final reaction, the **Mf** was obtained using EDC coupling, whereas the degrees of modification were ensured by the kinetically characterized TEMPO-oxidation. After the chemically modulated preparation of **Mf** had been performed, avidity characterizations were then undertaken.

### 2.3. Avidities of Oxidized Mannan Derivatives

In this study, human MBL (hMBL) and SP-D (hSP-D and pSP-D, for humans and porcine, respectively) were used as glycan-recognizing proteins. In order to estimate the avidities for these collectins, an ELISA-like assay was developed as already described [18]: studied proteins were grafted onto magnetic beads, and the presence of caught ligands were revealed using collectins conjugated to horseradish peroxidases (HRP) and tetramethylbenzidine (TMB) (Figure 6).

hMBL and SP-Ds were directly grafted on pre-activated NHS magnetic beads. Using the same proteins, HRP conjugates were also prepared, but it appeared that only hMBL-HRP was efficient as the secondary protein. Then, all the following results were obtained by using the HRP conjugate as a ligand detector. Arrays were performed using a decreasing amount of mannan solution, from 100 ng/mL to 0.78 ng/mL. The avidities between hMBL, hSP-D, pSP-D, and mannan (**Mannan**, **Mox** DO%) are displayed in Figure 7. As expected for hMBL (Figure 7A), **Mannan** binds efficiently to this protein as the system used reached its maximum possible signal at 3.1 ng/mL. Concerning oxidized mannan, the signal intensities decrease following the degree of oxidation since signals obtained with **Mox 8%** and **10%** are lower than those obtained with **Mannan**. Obviously, the oxidation of mannan has a direct and deleterious impact on mannan avidity for hMBL. A similar strategy was applied to the SP-Ds (Figure 7B,C). Unfortunately, the utilized secondary protein was once again hMBL-HRP for these assays because no effective ELISA-like procedure was successfully achieved when employing the SP-D-HRP complex. Thus, the results from these assays represent simultaneous glycans avidities for SP-D and hMBL. Fortunately, the involvement of hMBL had been previously described; thus, the actual avidities for SP-D were able to be estimated. Results from assays using hSP-D were gathered in Figure 7B. In comparison to the previous results with hMBL, the average signals were lower, likely due to different grafting efficiencies on magnetic beads. Even if the maximum avidity for **Mannan** was reached at 50.0 ng/mL, the trends obtained through oxidized mannan derivatives are globally more similar than in previous instances. Indeed, once again, the oxidation led to a decrease in avidity between the mannan and the protein. Thus, the involvement of mannuronic moieties seems to be equivalent for CRDs both from hMBL and hSP-D. Finally, avidity assays were performed using pSP-D (Figure 7C). **Mannan** is well recognized by pSP-D with a maximum signal obtained at 12.5 ng/mL, but the oxidation of mannan led to a different result than seen previously. The result obtained with **Mox 8%** was slightly lower than the one obtained with **Mannan**, indicating that in this experiment, the oxidation did not have a significantly deleterious impact on recognition. Although results are less blatant in the cases of **Mox 10%** and **Mox 24%**, this observation can also be made. Indeed, trends from **Mox 10%** and **Mox 24%** are lower than **Mannan**, but since hMBL was used as a secondary protein, losses of signal were expected. In fact, the results obtained with the oxidized mannan and pSP-D are at least two times higher than the results obtained with the fully hMBL-based assay. Thus, these assays suggest that the oxidation of mannan does not induce a loss of avidity for pSP-D.

### 2.4. Avidities of Fluorescent Mannan Derivatives

A similar protocol was utilized to survey the avidities of fluorescent mannan derivatives, which are represented in Figure 8. Fluorescent mannan **Mf**s, obtained from **Mox** 8 and 24% DO, were challenged. Surprisingly, for all proteins, results obtained with both **Mf**s were higher than those from the corresponding **Mox**. In the assays with pSP-D (Figure 8C), **Mf 8%** was even slightly higher than **Mannan**. Finally, the resulting fluorescent mannans were recognized by lectins, and their avidities were easily estimated according to the avidity of the utilized oxidized mannan.

### 2.5. Additional Studies with the Novel Ligands

EDTA-based assays were also used in order to ensure that described results were all based on the calcium complexation (Figure 9). Since no significant signal was obtained following the addition of EDTA, this assay revealed that all utilized mannan derivatives were based on the Ca^2+^ complexation mechanism.

A set of experiments with different concentrations of calcium was also used to determine the cause behind the loss of avidity between human MBL and **Mox**. Using the same ELISA-like protocol as described before, avidities between MBL, **Mannan**, and **Mox** were studied at different calcium concentrations (5, 30, and 100 mM) (Figure 10). The results showed that an increase in calcium concentration does not alter the avidity of MBL for **Mannan**. However, it does have a significant impact on the avidity of MBL for **Mox**, with an increase in signal at a Ca^2+^ concentration of 30 mM and above. The signal intensity obtained with **Mannan** could not be reached with **Mox** in spite of increased Ca^2+^ concentrations. These results showed that Ca^2+^ concentration impacts the avidity of MBL for **Mox** but is not the sole factor involved in the loss of avidity in comparison with **Mannan**.

We also investigated whether the novel fluorescent mannan was able to be used as a probe for immune cells. Thus, murine macrophages were exposed to solutions of **Mf 8%** and **Mf 24%** (1 mg/mL), and confocal microscopy imaging was performed (Figure 11). The results showed that **Mf** is well recognized and is caught by macrophages since a consequent number of fluorescent signals are observable in the cytoplasms.

### 2.6. Competitions between Mannan Derivatives

Investigations of two-stage competitions were undertaken, for which pre-bonded and lower avidity fluorescent mannan was supposed to be displaced by high-avidity ligands. In these assays, **Mf** 8% and 24% DO were first bonded to lectins grafted on magnetic beads. Because of their decreases in avidity for human lectins, their captures were achieved at higher concentrations (300 ng/mL) and were confirmed by measuring fluorescence emitting along flow cytometry analyses (Figure 12). After an exposition to glycans, the physical repartition does not undergo any modifications, meaning there is no aggregation phenomenon that could hinder the interpretations of results. Finally, fluorescence emitted from **Mf 24%** bonded to lectins was clearly observed.

Comparing distributions from **Mannan** and **Mf** (Figure 13), the delimitation of the positive fluorescent population (F+) was possible and used to quantify the remaining fluorescence from beads during competition assays.

The resulting fluorescent beads were then discharged in glycan solutions, and the remaining F+ population was compared to the initial F+ population after 60 min (Figure 14). Concerning the experiments involving hMBL (Figure 14A), the fluorescence from bonded **Mf 8%** was constant even after contact with concentrated solutions of **Mannan**. On the other hand, the resulting signal with bonded **Mf 24%** significantly decreased the following expositions to **Mannan** solutions. Despite triplicate-based assays, the sensitivity and repeatability of these results were as good as previous avidity studies. Similar results were obtained with assays using hSP-D (Figure 14B). Concerning assays using human proteins, **Mannan** was able to act as a substitute for formerly bonded **Mf** due to its higher avidity, and it appears that these substitutions are stimulated by the discrepancy between the respective avidity of each ligand. For assays with fixed pSP-D (Figure 14C), no loss of fluorescence was observed during contact with **Mannan** solutions, meaning that formerly bonded **Mf**s are stable in CRDs. Once again, the differences between MBL and SP-D are outlined here. Finally, the caught **Mf**s can be displaced when they come into contact with **Mannan** but only for human proteins.

### 2.7. Competition Assays Using Endotoxins

After proving that high-avidity ligands are able to release a pre-bonded ligand from lectins, the competition assays were challenged by using LPS. Three LPS from different strains were used: *Escherichia coli* O26:B26 (**LPS1**), O55:B5 (**LPS2**), and O111:B4 (**LPS3**). First, the avidity of each **LPS** was determined using the previous ELISA-like procedure (Figure 15). For these assays, **LPS1** and **LPS3** are both well recognized by all lectins, unlike **LPS2**. Where **LPS3** is the endotoxin with the highest avidity for hMBL (Figure 15A) and **LPS1** has the highest avidity for hSP-D (Figure 15B), the outcomes for pSP-D (Figure 15C) are less obvious since hMBL is used as a secondary protein. Indeed, even if the results obtained with **LPS1** seem to be lower than those with **LPS3**, this may be due to an underestimation resulting from the low avidity between **LPS1** and hMBL-HRP. Thus, **LPS1** and **LPS3** are both recognized ligands by human or porcine proteins. Unfortunately, the utilized system does not allow the detection of endotoxin in clinical situations (intervals of around 500 pg/mL) [21].

Based on previous competition assays using **Mannan**, a similar protocol was performed in order to check whether **LPS** were able to release pre-bonded **Mf 24%** from lectins. The outcomes are shown in Figure 16. As expected, highly concentrated **LPS1** and **LPS2** were unable to remove bonded **Mf 24%** from hMBL (Figure 16A). However, starting from 1 µg/mL, **LPS3** was able to displace bonded **Mf 24%** and induce a decrease in fluorescence from beads. Thus, despite the use of highly concentrated **LPS3** solutions, this result proved that competition between **LPS3** and **Mf 24%** occurs. Moreover, this effect was observed in the range of the ELISA-like protocol. Concerning assays with hSP-D (Figure 16B), despite a less stable signal due to a lower fluorescence signal being emitted from beads, none of the **LPS** were able to significantly release pre-bonded **Mf 24%** at these concentrations. On the other hand, **LPS1** and **LPS3** were able to induce a clear decrease in fluorescence from pSP-D-covered beads, despite **Mannan** not being able to release **Mf 24%** in the previous competition assays. Thus, according to the utilized protein, specific endotoxins have enough avidity for lectins to displace pre-caught fluorescent ligands.

## 3. Discussion

### 3.1. Preparation of the Fluorescent Ligands

In this study, the new concept of a two-stage competition was examined. A flexible and fluorescent ligand was first caught by collectins and then released at contact with high-avidity glycans. To achieve our objective, special attention was paid to the flexible ligand. To create it, we chose mannan as a starting material due to its high avidity and simple chemical composition. Then, the oxidation process of this polymer was undertaken in order to attain a kinetic tool to control the resulting avidity. Structural cleavages using acidic treatments were also considered. This oxidation step was also meant to implement carboxylic acids within the mannan structure, offering a way to chemo-selectively couple a fluorophore without modifying diequatorial vicinal hydroxyl groups. TEMPO catalysis recently made a breakthrough in the field of polysaccharide oxidation for its green aspect in the industrial preparation of nanofibers [26,27]. Spier et al. recently described the optimized conditions of polysaccharide TEMPO-oxidation, in which the polymer degradation is considerably reduced, but the DO is still useful [24]. The adapted protocol was successful since **Mox** was efficiently obtained without a significant loss of mass. The DO was able to reach 25%; however, thanks to the kinetic study, a lower DO can easily be obtained (such as with **Mox** 8% and 10%). The results showed that no significant degradation of the original polymer occurred. This mass conservation is in good agreement with the literature since mannan from *Saccharomyces cerevisiae* is essentially based on α(1-2) and α(1-6) linkages [28], and the main degradation mechanism under oxidation is usually attributed to β-elimination [29]. Thus, the TEMPO-oxidation process appeared to be smooth and efficient in order to prepare **Mox** for specific DO.

In order to obtain the fluorescent mannan, we next focused our attention on the fluorophore fragment. Due to their high fluorescence intensity [30] and their numerous applications across biological studies [31,32], fluorescein derivatives caught our attention. The synthesis pathway was elaborated in order to implement a terminal amine function and a spacer in the fluorescein structure. The proposed synthesis strategy begins with the formation of the methyl ester fluorescein to avoid unwanted substitutions from the carboxylic acid during the subsequent Williamson coupling. Another point was also considered in concordance with the functioning principle of CRDs: since the recognition of mannan by these proteins is based on Ca^2+^ coordination, we were afraid that the incorporation of fluorophore-bearing carboxylic acid moieties could interfere with avidity studies. Thanks to this protocol, if necessary, the saponification can be bypassed to reach an esterified version of **3**, which would have fewer interactions with cations. Nevertheless, according to the avidity studies, it appeared that **Mf**s have higher avidities than their respective **Mox**, meaning that the carboxylic acid of fluoresceins has not a deleterious impact. Even if the proposed protocol leads to a significant decrease in fluorescence in comparison with native fluorescein, **Mf**s were successfully obtained and were still able to be used as detectable fluorescent probes (Figure 11). Based on this synthetic pathway, different fluorophores may also be grafted, thus paving the way for multi-fluorescent combinations of **Mf**s caught by different proteins theoretically may allow simultaneous monitoring of different ligands.

### 3.2. Effect of the Oxidation on the Recognition of Mannan by Collectins

The avidities assays outlined different trends concerning the interactions between collectins and the new derivatives of mannan. First, the oxidation of mannan has a direct and deleterious impact on mannan avidity for human proteins specifically. This result was observed by using the common ELISA-like procedure but also by using the novel competition-based assays, where **Mannan** was not able to displace **Mf**s caught by pSP-D. The additional experiments with a higher concentration of calcium showed that mannuronic acid moieties interact with the cation, preventing recognition by human lectins. Nevertheless, another effect is mainly responsible for the loss of avidity between **Mox** and MBL. This effect seems to be specific to human proteins, thus explaining why pSP-Ds were able to effectively catch **Mox**, despite the interaction between calcium and mannuronic acid moieties. This effect might be due to conformation changes within the oxidized mannan structure, as has been described in SEC analyses. Only hypotheses can be drawn here since the involvement of mannuronic moieties in lectins recognition is still uncertain and requires further study. Different studies have already described the discrepancies between both species: porcine SP-D is known for having a broader range of Influenza A virus recognition than human lectins [33]. Their immunologic discrepancies were attributed to differences within their respective structures: pSP-Ds bear a sialylated oligosaccharide moiety and an extra peptidic sequence known as the “GSS-loop” [34,35]. Nevertheless, this is the first time that the involvement of carboxylic acids at a C6-position is outlined. Further and specific studies are required to understand the issues behind this effect. However, as the purpose of this study is to create fluorescent ligands with reduced avidities, this effect perfectly fits, and it was effectively used to highlight the two-stage competition.

### 3.3. Recognition of the Fluorescent Ligands and Endotoxins

The demise of the mannuronic moieties after the amide formation with the fluorescein derivative **4** reduces this deleterious effect on avidity, explaining why **Mf**s have higher avidities than their corresponding **Mox**. Nevertheless, **Mf** 8% displayed a slightly higher avidity than the native **Mannan** with pSP-D. This result causes us to consider that the avidity was also improved by structure unraveling and the unleashing of hindered sites, following the inclusion of aromatic moieties within the mannan structure. Ultimately, **Mf** was caught by the different collectins, and the fluorescence emitted from the beads was clearly observed. On the other hand, endotoxins were hardly recognized by our system. Indeed, our interval of detection was considerably higher than the concentration of PAMPs in clinical situations (at around 500 pg/mL) [21]. Even if the proposed system is fairly similar to the one described by Cartwright et al., it was not able to reach the sensitivity they achieved (around 15 ng/mL) [18]. This issue is probably due to the protein used and the process of fixation. Indeed, in their study, an engineered MBL was used (FcMBL), and their tethering of magnetic beads was ensured by a biotin–streptavidin complexation. The proposed system here would probably lead to several modifications of the oligomeric states of the proteins. The efficiency of collectins mainly hinges on their oligomeric states, allowing multivalent interactions [36,37]. The association of SP-D with HRP also leads to the critical modifications of the oligomeric states and the loss of multivalent effects, which makes them inefficient as secondary proteins. Since the purpose of this study was to highlight the concept of a two-stage competition, no due consideration was given to the protein and the tethering process. The main objective was successfully achieved since **Mf**s are recognized by lectins, and their chemically crafted avidity ensures their replacement by a highly recognized ligand.

### 3.4. Two-Stage Competitions

Competitions assays were performed by analyzing the remaining fluorescence from beads after the exposition of highly recognized ligands. The results obtained with **Mannan** were coherent with the avidity studies. Indeed, **Mannan** was only able to displace **Mf 24%** caught by human proteins. These results were expected since this ligand displays the lowest avidity regarding these proteins. The fluorescence stabilization of beads bearing pSP-D was also expected since the results showed that chemical modifications within the mannan structure do not induce any avidity discrepancy for this protein. Thus, this assay proved that a ligand with equal avidity was not able to remove a formerly bonded ligand from pSP-D. According to the results with human proteins, competitions between **Mannan** and pre-bonded **Mf** indeed occur, proving that a higher avidity glycan can replace previously bonded glycan in collectin mechanisms. Nevertheless, this competition depends on the collectins used and the gap between the respective avidity of both glycans. In our case, the decrease in fluorescence after the exposition of **Mannan** was ensured by the modulate decrease in the avidity of **Mf** following the monitored TEMPO-oxidation. By using such protocol and human recombinant proteins, the presence of **Mannan** could be detected after sixty minutes of incubation and without using any secondary proteins. Despite the lack of sensitivity, our system proved that LPS could also displace pre-bound **Mf** from lectins. Indeed, the MBL-based systems were able to detect **LPS1**. The pSP-D-based experiments showed a decrease in fluorescence in contact of **LPS1** and **LPS3**, even though **Mannan** was unable to displace **Mf**. Thus, two-stage competitions occur, but only a recognized LPS (or glycan) is able to remove the fluorescent ligand from the lectins. Moreover, it also appeared that the unrecognized ligands (like **LPS2**) did not impact fluorescence signals, meaning that the developed system seems to be extremely selective. This advantage would ensure long-term interactions in cases of exposition with unrecognized glycans and would enable the avoidance of any false positives, which is a common issue in such analyses [38]. In a general manner, the sensitivities of the two-stage competition assays were similar to that obtained through the classic ELISA-like protocol. Then, by improving the ELISA-like protocol, this new system should also be optimized. This system also fits with the multiplex array beads: a combination of fluorescent mannan caught by different lectins may allow the detection and the identification of endotoxins. For example, a mixture of supported hMBL and pSP-D, identifiable through two different caught fluorescent mannans, can theoretically allow the detection and the determination of **LPS1** or **LPS3** (both supports lose their fluorescence following an exposition to **LPS1**, but only the beads bearing pSP-D lose theirs when in contact with **LPS3**). Finally, by characterizing a large quantity of optimized collectins, such a competitive system associated with the modulate preparation of **Mf** allows the creation of a diagnostic tool suitable for the multiplex bead array assay.

## 4. Materials and Methods

### 4.1. Materials

All the reagents and solvents were obtained from commercial sources and used without further purification. All the chemical reactants were provided by Sigma-Aldrich (St. Louis, MO, USA), and all the biochemical consumables were obtained from Fisher Scientific, including 3.5 kD SPECTRA/POR 7 Spectrum-Labs RC dialyses tubes, activated NHS-magnetic beads and LPS solution 500X from *Escherichia coli* O26:B26 (Invitrogen, Waltham, MA, USA). Mannan from *Saccharomyces cerevisiae* was supplied by Carbosynth. Human Mannan Binding Lectin 2 (hMBL, Sinobiological) and the Lightning Kit HRP were provided by Interchim. Recombinant hSP-D and pSP-D were produced using HEK293E cells and purified by mannan–agarose affinity chromatography, as described in detail previously [35,39,40]. All tris-buffered saline–tween (TBST) Ca^2+^ buffers were prepared by including 5 mM of calcium chloride in TBS (1X) with 0.1% Tween-20. Other LPS samples from strains O55:B5 and O111:B4 were obtained as dried powders from Sigma-Aldrich.

### 4.2. Instrumentation

The 1H and 13C nuclear magnetic resonance (NMR) spectra were recorded on a Bruker AV1 300 spectrometer (Bruker BioSpin GmbH, Rheinstetten, Germany) working at 300 MHz, 75 MHz, respectively, for 1H and 13C, with deuterated solvents. Chemical shifts were reported in δ, parts per million, downfield from internal TMS. Coupling constants, *J*, were reported in Hertz (Hz) and refer to apparent peak multiplicities and not true coupling constants. The abbreviations s, d, dd, t, q, br, and m stand for resonance multiplicities singlet, doublet, doublet of doublet, triplet, quartet, broad and multiplet, respectively. High-resolution mass spectrometry data were recorded with an accuracy of within 5 ppm on a quadrupole-TOF mass spectrometer (Xevo Q-Tof, Waters, Guyancourt, France) using an electrospray ionization source operating in positive mode. Thin-layer chromatography (TLC) was carried out on aluminum sheets pre-coated with silica gel plates (Fluka Kiesel gel 60 F254, Merck, Bucharest, Romania) and visualized using a 254 nm UV lamp. SEC-MALS analyses were performed by the “Plateforme Interactions des Macromolécules” (PIM) from the “Institut de Biologie Intégrative de la Cellule” (I2BC, Orsay, France) using their HPLC (Shimadzu, Kyoto, Japan), LS detector (miniDAWN TREOS, Wyatt Technology, Santa Barbara, CA, USA) and RI detector (T-rEX, Wyatt Technology). Absorption spectra were recorded on a PerkinElmer Lambda 750 UV/VIS/NIR spectrometer (PerkinElmer, Haguenau, France). Fluorescence spectra were recorded on a Varian Cary Eclipse Spectrophotometer (Varian, Le Plessis-Robinson, France). All the spectra were recorded from solutions introduced in four-face quartz cuvettes. Fluorescence quantum yields (Φf) were determined using a fluorescein solution in NaOH 0.1 M as the fluorescence standard (Φf = 0.95). Tap water was distilled with Millipore RiOs and Synergy UV Purification Systems (Merck, Molsheim, France). IR spectra were performed with a Thermo Scientific Nicolet 6700 FT-IR Spectrometer (ThermoFisher, Illkirch, France). Conductimetric titration and pH monitoring were performed using Fisherbran accumet AB200 pH/Conductivity Benchtop Meters (ThermoFisher, Illkirch, France). Samples were freeze-dried inside a Christ Alpha 2–4 LSCbasic (Grosseron, Couëron, France). Avidities and competitions assessments were performed using magnetic beads operated by a Thermo Scientific KingFisher Duo Prime purification system (ThermoFisher, Illkirch, France). For ELISA-like avidity array assays, the absorbance of oxidated TMB solutions were determined by measuring absorbance at 450 nm with a BMG LABTECH Fluostar OMEGA Plate Reader (BMG Labtech, Champigny-sur-Marne, France). For competition assays, fluorescence emitting from beads was analyzed using a BD LSR Fortessa Cell Analyser (BD Biosciences, Le Pont de Claix, France). The murine macrophages (J774.2) were cultured in Dulbecco’s Modified Eagle Medium (DMEM), containing 5% fetal bovine serum (FBS) and 1% penicillin–streptomycin (P/S). Cells were examined by confocal microscopy under a WLL Leica SP8 Microscope, using a FITC filter.

### 4.3. General Procedure for the TEMPO-Oxidation of Mannan (***Mox***)

A solution of mannan was prepared by dissolving 50 mg of yeast manna into 20 mL of pure water by stirring overnight at room temperature. The solution of mannan and 27 mL of pure water was added into a three-neck round bottom flask containing (2,2,6,6-tetramethylpiperidin-1-yl)oxyl (TEMPO) (7.5 mg, 48 µmol) and sodium bromide (3.9 mg, 38 mmol) under stirring and placed in a cold bath. pH monitoring was set up by introducing a probe while paying attention to the magnetic bar. A sodium hypochlorite solution (11%, 3.0 mL, 396 mg, and 3.3 mmol) was added dropwise to the cold solution while stirring. Quickly, a few quantities of HCl 0.1 M were added in order to decrease pH to 9.3, which is considered to be the starting point for the kinetic study. Using pH monitoring, pH was maintained between 9.0 and 9.3 by frequently adding droplets of 0.1 M solution of NaOH. For the kinetic study, samples were extracted and quenched by adding NaBH_4_ dissolved in methanol (1.5 mg/mL). After 2 h, pH was no longer evolving, and 20 mL of NaBH_4_/methanol was added. The batch was directly dialyzed against distilled water and freeze-dried to obtain **Mox** with 90% of the initial mass. Conductometric titrations were performed to measure the carboxylic acid concentration: 30 mg of **Mox** were dispersed into a 0.01 M HCl solution and shaken for 10 min. The conductivity of this solution was recorded during the progressive addition of a 5 mM NaOH solution. The carboxylic acid content was calculated as previously described [25].

### 4.4. Synthesis of Fluorescein Derivative and Fluorescent Mannan

The synthesis of methyl ester fluorescein (**2**) was performed in a round bottom flask containing fluorescein (**1**) (0.333 g, 1 mmol), and dry methanol (35 mL) was added dropwise to a solution of concentrated sulfuric acid (0.35 mL, 6.3 mmol and 6.3 eq). The mixture was refluxed overnight at 80 °C under an inert atmosphere. The resulting suspension was then cooled, and the solvent was removed under vacuum. The product was precipitated in cold water, then filtered on a Buchner funnel, and finally washed and dried to obtain a brown powder with a quantitative yield. ^1^H NMR (300 MHz, dimethylsulfoxide (DMSO)_d6_): δ (ppm) = 3.56 (s, 3H, OMe), 7.09 (dd, 2H, *J* = 9.5 and 2.0 Hz, H_o-xan_), 7.23 (d, 2H, *J* = 2 Hz, H_o-s-xan_), 7.30 (d, 2H, *J* = 9.5 Hz, H_m-xan_), 7.55 (d, 1H, *J* = 7.6 Hz, H_δ-COOMe_), 7.88 (t, 1H, *J* = 7.7 Hz, H_β-COOMe_), 7.96 (t, 1H, *J* = 7.6 Hz, H_γ-COOMe_), 8.31 (d, 1H, *J* = 7.7 Hz, H_α-COOMe_). HRMS (ESI+) [C_21_H_14_O_5_]: *m/z* [M+H]^+^ calculated 347.0841, found 347.0906.

The synthesis of N(Boc)-hexyl fluorescein (**3**) was carried out in a round bottom flask containing **2** (0.346 g, 1 mmol), dry N,N-Dimethylformamide (33 mL), K_2_CO_3_ (1.380 g, 10 mmol, 10 eq) and 6-(Boc-amino) hexyl bromide (0.388 g, 1.4 mmol, 1.4 eq). The suspension was refluxed for a night at 70 °C under an inert atmosphere. After the removal of the solvent under vacuum, the mixture was washed with ethyl acetate and water. The aqueous layers were washed with ethyl acetate, and the organic layers were collected and dried on MgSO_4_. The crude was directly dissolved in a solution of methanol (16 mL) and LiOH 1 M (7.8 mL). The mixture was stirred overnight at room temperature. The suspension was then concentrated, neutralized with 1 M HCl (7.8 mL), and decanted using ethyl acetate and water. The aqueous layers were washed with ethyl acetate, and the organic layers were collected and dried on MgSO_4_. The final product was purified by silica gel chromatography (95 dichloromethane/5 methanol) to obtain a yellow solid with a 70% yield. ^1^H NMR (300 MHz, Me_d_OD): δ (ppm) = 1.35–1.59 (m, 17H), 3.05 (t, 2H, *J* = 6.7 Hz, H_α-NHBoc_), 4.03 (t, 2H, *J* = 6.3 Hz, H_α-OFluo_), 6.50–6.71 (m, 5H, H_xan_), 6.85 (s, 1H, H_m-Ohexyl_), 7.21 (d, 1H, *J* = 7.6 Hz, H_δ-COO_), 7.71 (t, 1H, *J* = 7.5 Hz, H_β-COO_), 7.78 (t, 1H, *J* = 7.6 Hz, H_γ-COO_), 8.03 (d, 1H, *J* = 7.4 Hz, H_α-COO_). HRMS (ESI+) [C_31_H_33_NO_7_]: *m/z* [M+H]^+^ calculated 532.2335, found 532.2342.

The synthesis of amino-hexyl fluorescein (**4**) was performed in a round bottom flask containing **3** (0.183 g, 0.35 mmol), and a 4 M solution of HCl/dioxane (6.88 mL) was added dropwise while stirring. The solution was kept under stirring for 30 min. The product was diluted into pure water and then concentrated under vacuum three times to remove HCl. The aqueous solution was finally removed, and compound **4** was freeze-dried to obtain a yellow powder with a quantitative yield. UV/Fluo (NaOH 0.1 M): ΦF(exc: 475 nm) = 0.32. ^1^H NMR (300 MHz, Me_d_OD): δ (ppm) = 1.39–1.96 (m, 8H), 2.96 (t, 2H, *J* = 7.1 Hz, H_α-NH3_), 4.17 (t, 2H, *J* = 6.2 Hz, H_α-OFluo_), 6.73–7.00 (m, 5H, H_xan_), 7.11 (s, 1H, H_m-Ohexyl_), 7.30 (d, 1H, *J* = 7.4 Hz, H_δ-COO_), 7.78 (t, 1H, *J* = 7.4 Hz, H_β-COO_), 7.84 (t, 1H, *J* = 7.4 Hz, H_γ-COO_), 8.15 (d, 1H, *J* = 7.4 Hz, H_α-COO_). HRMS (ESI+) [C_26_H_25_NO_5_]: *m/z* [M+H]^+^ calculated 432.1811, found 432.1797.

The general procedure for preparation of fluorescent mannan (**Mf**) was carried out in a round bottom flask containing **Mox** (15 mg, 8% DO = 7 µmoL to 25% DO = 21 µmol, 1 eq), in which EDC (1.5 eq), NHS (1.5 eq) and 5 mL of MES Buffer pH 4.7 were placed. After 5 min, dissolved **5** (1.2 eq) in 100 µL of DMSO was slowly added to the solution of **Mox** under stirring. The solution was stirred for 2 days at room temperature. The fluorescent polysaccharide was purified by dialyzing against distilled water and freeze-drying to reach the final **Mf** as a light yellow solid with a 50% mass conservation.

### 4.5. Preparation of Magnetic Beads and Avidity Assays

HRP conjugates, such as hMBL-HRP, were prepared by using the Lightning-Kit HRP Conjugation kit and 50 µg of lectins (hMBL or SP-D), according to the recommended procedure. Lectin-covered particles were prepared by coupling 25 µg of lectins (hMBL or SP-D) with 1 mg of NHS-activated magnetic beads following the recommended automatic procedure for protein coupling on the KingFisher DuoPrime system. Final bead suspensions were prepared in TBST Ca^2+^ (5 mg/mL) and stored at 4 °C for two months maximum. All polysaccharide solutions were prepared by dissolving freeze-dried samples in TBST Ca^2+^ 5 mM. Avidity assessments were achieved according to the following steps: 2.5 µg of beads were gently mixed in 1 mL of polysaccharides solution for 20 min at room temperature. Beads were washed three times in 500 µL of TBST Ca^2+^, using a magnetic support. Beads were then dropped into 500 µL of HRP conjugate solutions (50 ng/mL in TBST Ca^2+^) and mixed for 20 min at room temperature. Beads were washed three times in 500 µL of TBST Ca^2+^, then dropped into 100 µL of TMB solution (1X) and gently mixed for 8 min. After bead retrieval, 50 µL of sulfuric acid solution (1 M) was added to the resulting solution. Samples transferred into NUNC 96-well plates were analyzed in a plate reader by measuring the absorption at 450 nm. For EDTA-based assessments, EDTA 10 mM was included in all TBST Ca^2+^ buffers, and the mannan concentration used was 250 ng/mL. For experiments with increasing concentrations of calcium, TBST Ca^2+^ 5 mM was replaced by TBST Ca^2+^ 30 mM or 100 mM. Avidity assessments were carried out as before, but 5 µg of beads were used.

### 4.6. Competitions between Fluorescent Mannan Derivatives and Others Glycosyl Structures

Competition assays were performed according to the following steps: 2.5 µg of beads were gently mixed in 1 mL of **Mf** (300 ng/mL) for 20 min at room temperature. Beads were washed three times in 500 µL of TBST Ca^2+^. Beads were dropped into 1 mL of polysaccharide solution (TBST Ca^2+^ for control, mannan, or lipopolysaccharides) and mixed for 60 min at room temperature. Beads were washed three times in 500 µL of TBST Ca^2+^ and dropped into 100 µL of TBS Ca^2+^. Twenty-fold diluted suspensions in TBS Ca^2+^ were analyzed by flow cytometry. Fluorescence emissions were analyzed following Alexa Fluor 488 filters, in which F+ and F- populations were dissociated using unexposed magnetic beads. For each condition, three independent assays were carried out using 20 k recorded events.

## 5. Conclusions

Accelerating accurate diagnostic analyses is one of the main focuses for reducing the consequences of global infections. Recently, multiplex beads array assays proved their ability to rapidly titrate several analytes in biological fluids. In this study, we investigated a two-stage competitive concept in order to detect highly recognized ligands, such as PAMPs. After describing a new preparation of fluorescent mannan through a TEMPO-oxidation, these ligands were studied for their specific avidities for collectins, especially hMBL and SP-D. The results showed that the implementations of mannuronic moieties within the structure of mannan induce a loss of avidity for human proteins but not for porcine SP-D. Despite this effect, these new fluorescent ligands were used in novel two-stage competitive assays, whereas highly recognized ligands unleash pre-bonded fluorescent mannan. The detection of recognized glycans was then achieved with good selectivity, repeatability, and sensitivity for mannan. The sensitivity of the new concept is similar to that of our ELISA-like procedure. A lack of sensitivity for PAMPs was described as being most likely due to the unexpected damaged of the oligomerization states, leading to turbulence within multivalent interactions. In comparison with the previous ELISA-like protocol, this novel process offers several advantages, such as the reduction in protein consumption and the saving of time, even if it requires a specific analytical device, such as a flow cytometer. The creation and characterization of modulate fluorescent mannan are a powerful chemical tool for the implementation of competitions in collectins and for the creation of diagnostics kits for the detection of high avidity glycans.

## Figures and Tables

**Figure 1 ijms-23-16067-f001:**
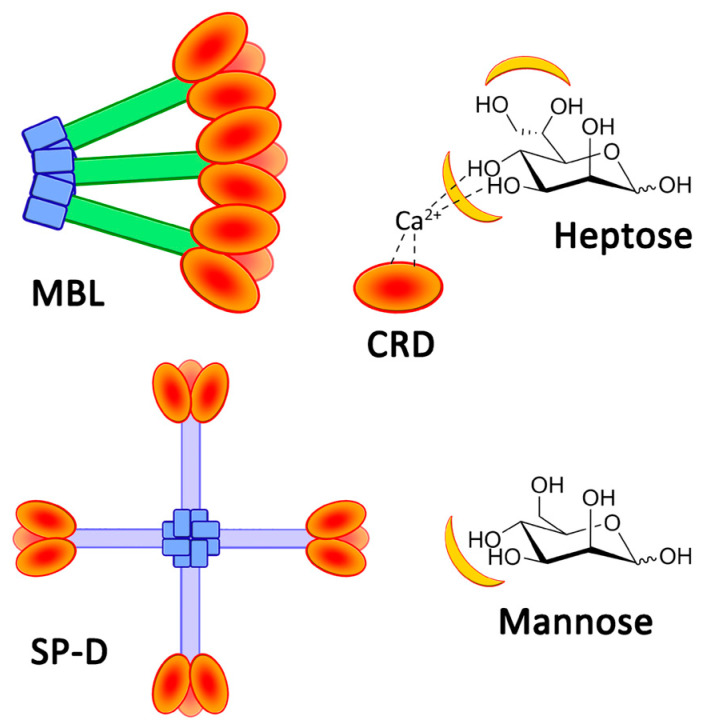
Representation of MBL and SP-D. The chemical structures of heptose and mannose, with lectin-recognized sites in yellow.

**Figure 2 ijms-23-16067-f002:**
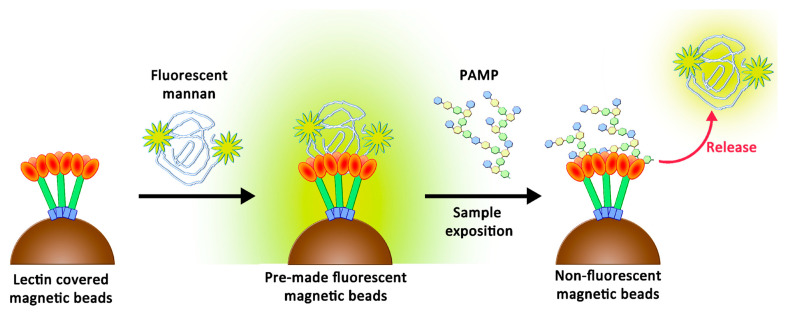
Two-stage competition between PAMPs and fluorescent mannan formerly caught by lectins.

**Figure 3 ijms-23-16067-f003:**
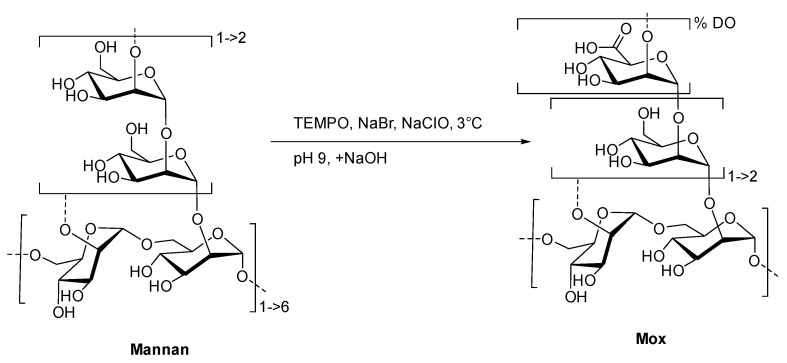
Formation of **Mox** following TEMPO-catalyzed oxidation.

**Figure 4 ijms-23-16067-f004:**
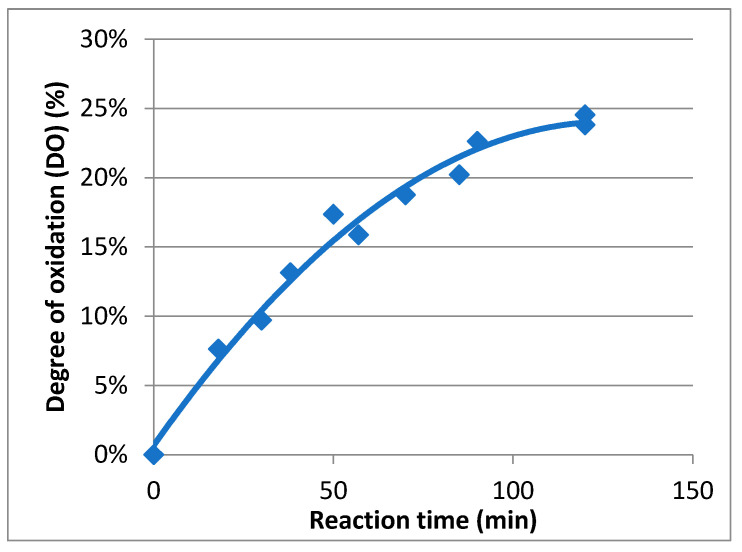
Kinetic study of the oxidation of mannan.

**Figure 5 ijms-23-16067-f005:**
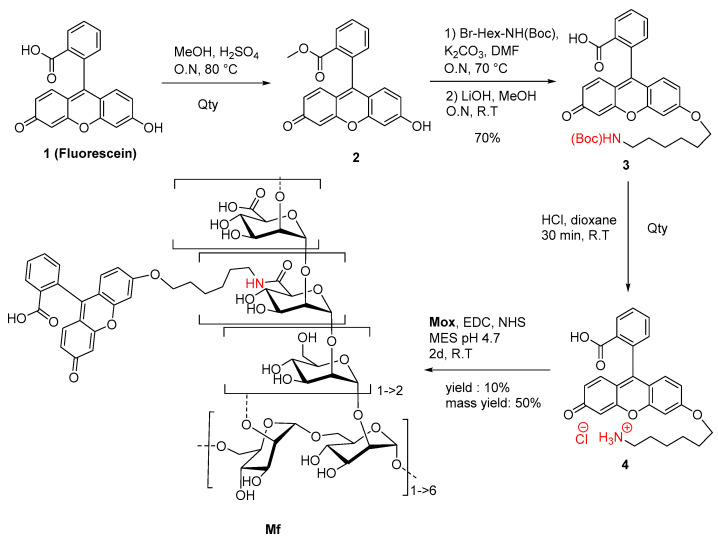
Synthetic route to **Mf**.

**Figure 6 ijms-23-16067-f006:**
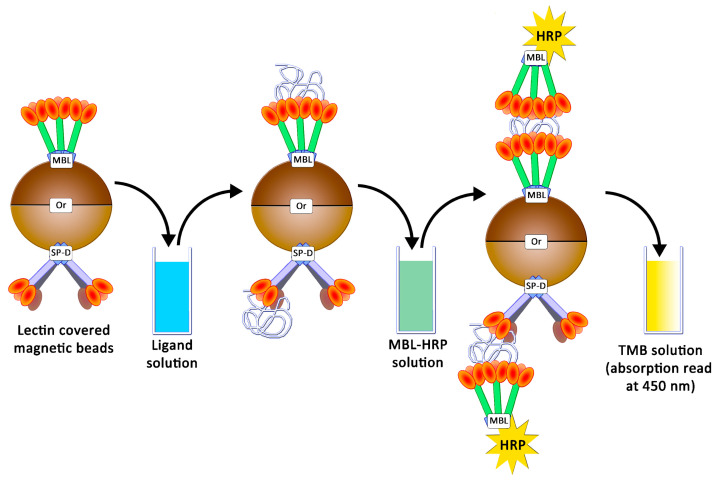
Avidity determination between ligands and lectins following an ELISA-like protocol.

**Figure 7 ijms-23-16067-f007:**
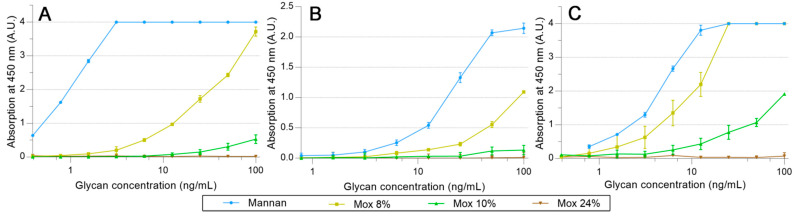
Avidity determination between oxidized mannan and fixed hMBL (**A**), hSP-D (**B**), and pSP-D (**C**). Lectins fixed on magnetic beads are mixed in mannan solutions for 20 min at room temperature. After three washing steps, beads are placed in a solution of hMBL-HRP for 20 min at room temperature. Avidity determination occurs after the oxidation of the TMB solution by washed beads and absorbance reading at 450 nm.

**Figure 8 ijms-23-16067-f008:**
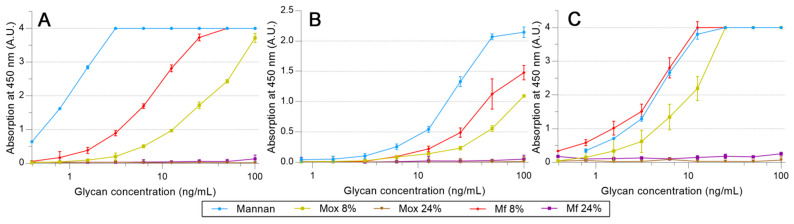
Avidity determination between fluorescent mannan and fixed hMBL (**A**), hSP-D (**B**), and pSP-D (**C**) obtained using the protocol described previously.

**Figure 9 ijms-23-16067-f009:**
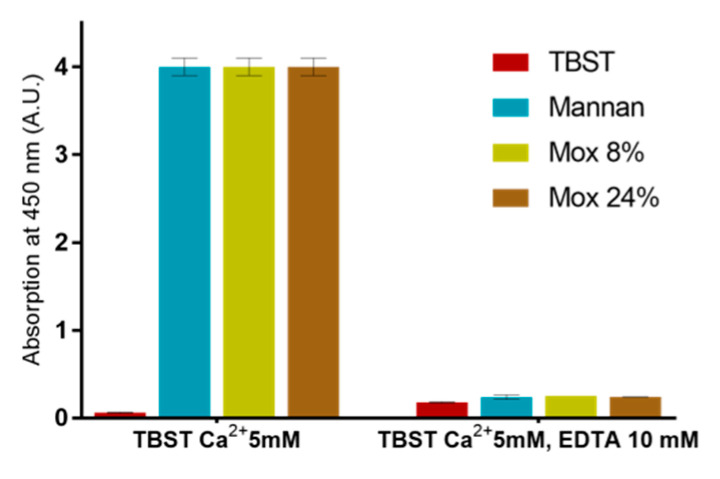
Calcium involvement in mannan recognition using EDTA-based assessments with different mannan at 250 ng/mL and MBL.

**Figure 10 ijms-23-16067-f010:**
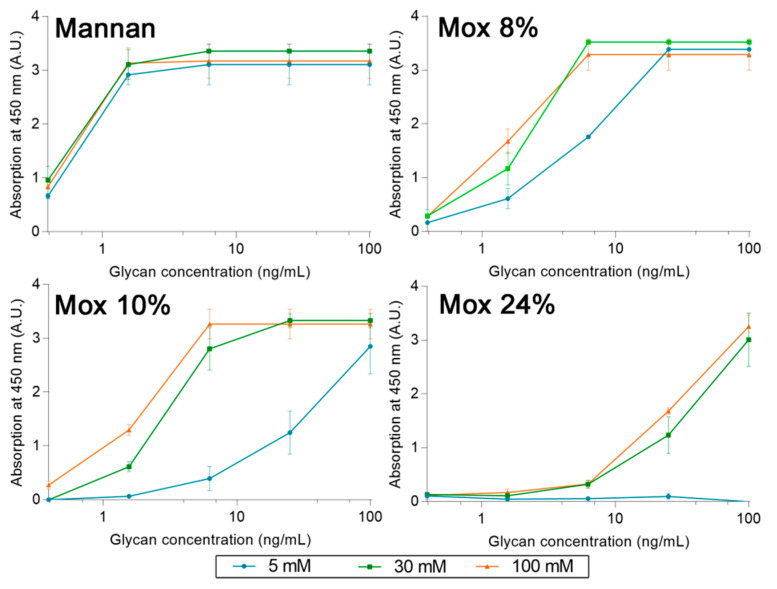
Avidities trends between **Mannan**, **Mox**, and human MBL, obtained using the protocol described before but with different concentrations of calcium chloride (5, 30, and 100 mM).

**Figure 11 ijms-23-16067-f011:**
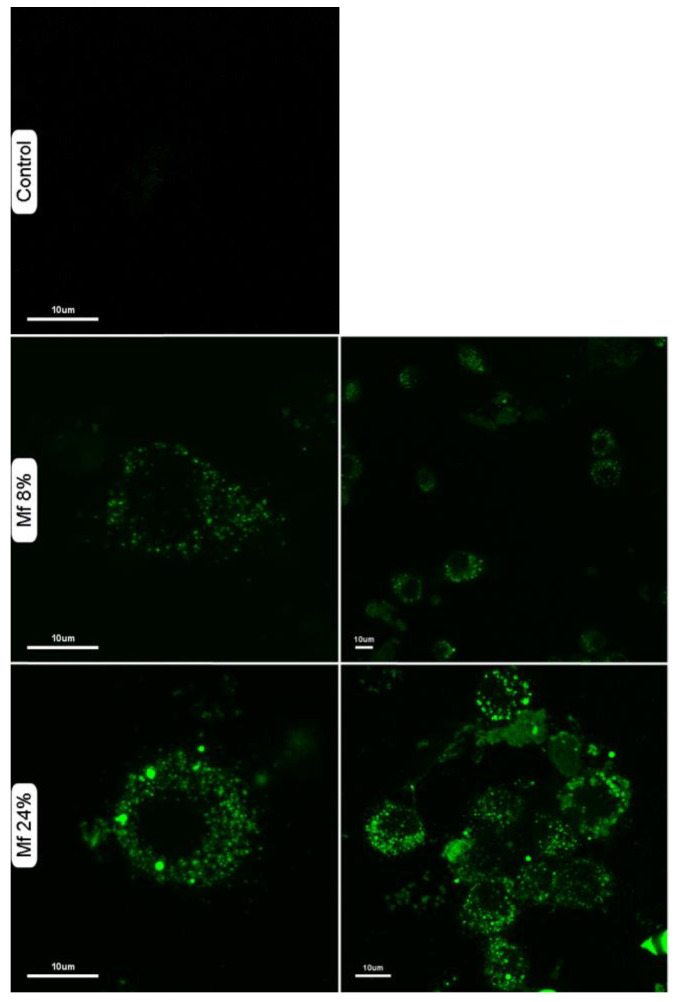
Confocal laser scanning microscopy imaging of murine macrophages in different conditions: untreated (control), after incubation with **Mf 8%** or **Mf 24%**. Macrophages were suspended in a solution of fluorescent mannan (1 mg/mL) for 30 min at 37 °C, then washed three times with buffer.

**Figure 12 ijms-23-16067-f012:**
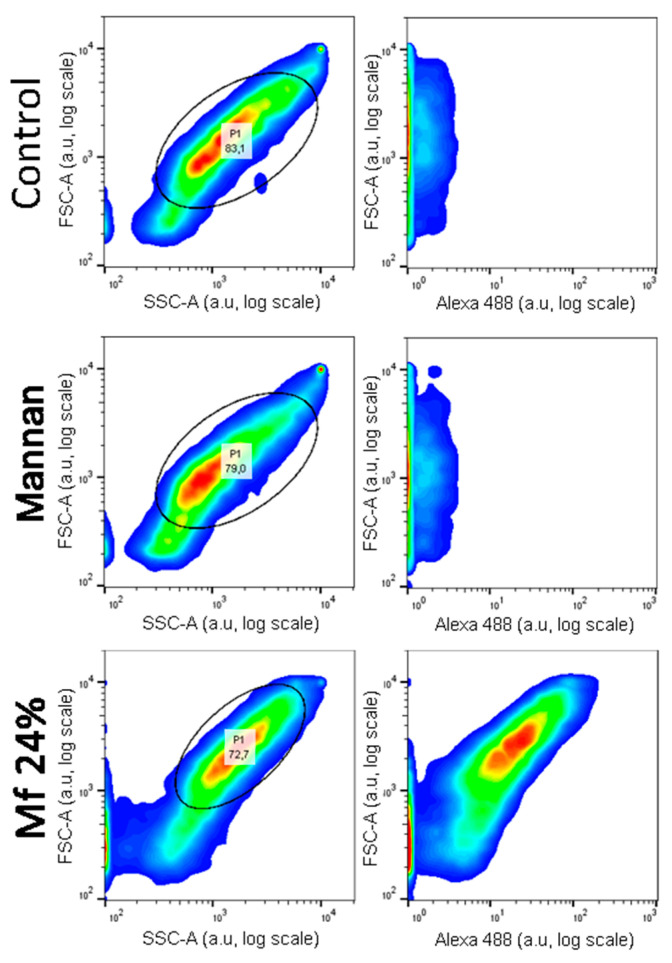
Flow cytometry analyses of hMBL-covered beads after 20 min in TBST Ca^2+^ buffer (Control), **Mannan** (100 ng/mL), and **Mf 24%** (300 ng/mL) solutions.

**Figure 13 ijms-23-16067-f013:**
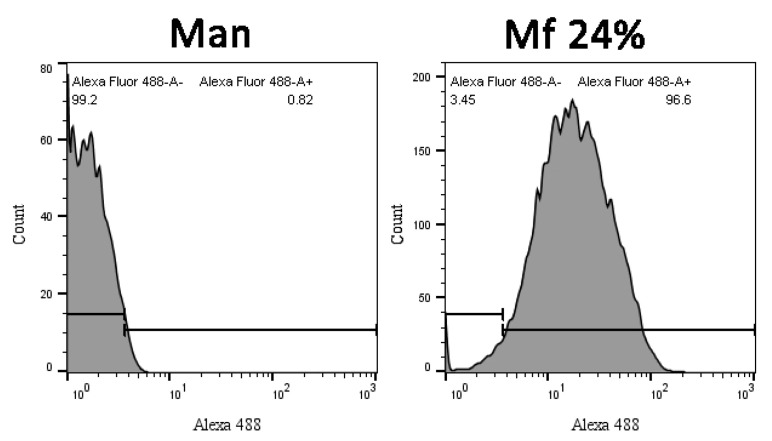
Cytograms of **Mannan** and **Mf 24%** bonded beads according to their fluorescence intensity.

**Figure 14 ijms-23-16067-f014:**
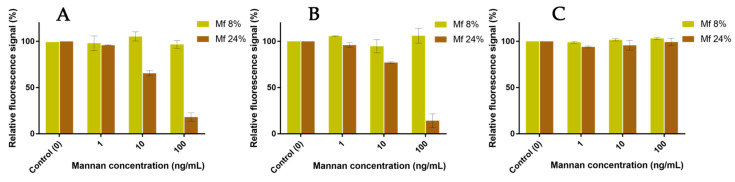
Competition assays of **Mf** (8% and 24%) bonded to fixed hMBL (**A**), hSP-D (**B**), and pSP-D (**C**) versus **Mannan** at different concentrations (1, 10, 100 ng/mL).

**Figure 15 ijms-23-16067-f015:**
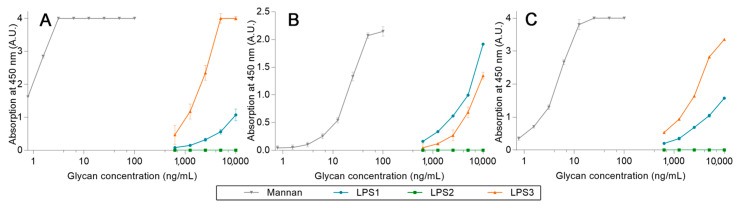
**LPS** avidity array assays using fixed hMBL (**A**), hSP-D (**B**), and pSP-D (**C**). Lectins fixed on magnetic beads are mixed in LPS solutions for 20 min at room temperature. After three washing steps, beads are placed in a solution of hMBL-HRP for 20 min at room temperature. Avidity determination occurs after the oxidation of the TMB solution by the washed beads and absorbance reading at 450 nm.

**Figure 16 ijms-23-16067-f016:**
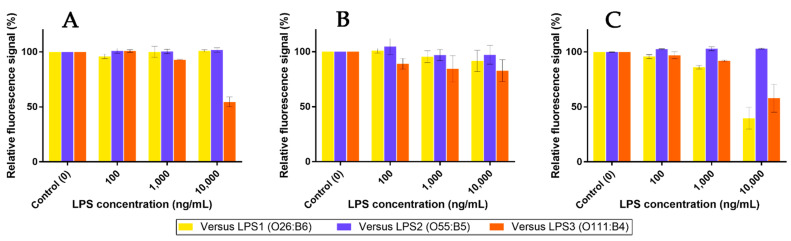
Competition assays of **Mf 24%** bonded to fixed hMBL (**A**), hSP-D (**B**), and pSP-D (**C**) versus **LPS** at different concentrations (0.1, 1, and 10 µg/mL).

## Data Availability

Data available on request from florent.le-guern@uvsq.fr.

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
