# Peer review of "Labeled TEMPO-Oxidized Mannan Differentiates Binding Profiles within the Collectin Families"

_ijms, 2022, doi:10.3390/ijms232416067_

Round 1
Reviewer 1 Report
The manuscript describes the use of oxidized mannans as low affinity ligand for human lectins in order to detect higher affinity ligands, such as bacterial glycans, with possible interest in diagnosis of sepsis. The authors were successful in the development of the methodology, but they could not achieve sufficient sensitivity for the detection of low concentration of endotoxin.
The concept and technical parts are well described, with some details missing in the methodology. The use of a “lower affinity” ligands for displacement assays is an interesting strategy. However, the sensitivity appears to be too low for application as biosensors. In my opinion, the authors did not consider the fact that is very difficult to displace multivalent interactions (mainly due to rebound effect), even with higher affinity ligand. The success of fluorescence polarization that uses monovalent ligands illustrates this. It would be of interest to evaluate if the beads format could be used with monovalent fluorescent derivative of mannose, maybe resulting in more efficient displacement (not asked by the reviewer in revision of article). On the other hand, deeper investigations in the mechanism of reduction of affinity for mannuronic acid would be of interest.
Suggested corrections (major ones first)
1. Some more investigations about the molecular mechanisms for lower affinity for Mox would have been of interest. The authors rose the question of possibility that lower binding of Mox (compared to Man) could be due to calcium trapping. It is important to clear this point since it may affect interpretation of data when using biological sample. It is recommended to test the effect of addition of increasing added calcium in the solution for both Mannan and Mox binding to lectins.
2. Also in the interest of better understanding the rational for lower affinity (and differences between human and porcine SPS), a measurement of affinity between lectins of interest and mannuronate (or oligomannuronate) could be performed.
3. The discussion bottom of page 13 about the concentration of PAMPS is serum and the limit for detection should be moved in the introduction since it is a prerequisite of interest.
4. The conclusion should be revised taking into account the problem of using multivalent ligands. The term “common proteins” is not clear and there is little hope that changing the lectin will overcome the problem of sensitivity.
5. In description of “Materials” – the indication that SPDs were “constructed” as published is too concise. A short description is requested (presence of tag ? purification procedure ?).
6. Due to the importance of multivalent effects in this study, a characterization/verification of the oligomeric states of the proteins would be useful.
7. Middle of page 2, the references (9-15) that describe a variety of experiments are grouped, while the sentence is only about MBP.
8. Some corrections of nomenclature. While “heptose” or Hep is a largely used name, it has to be stated as first use that it stands for “L-glycero-D-manno-heptose”. “Man” should not be used as abbreviation of mannan since it is the official abbreviation of Mannose. I would suggest to keep “mannan” as it is. The expression “a(1-2) and a(1-6) backboned derivatives” should be rephrased.
9. Figure 7 and 8 should be merged. Also, the x-axis could be represented starting from 0 (increasing concentrations, instead of decreasing ones).
10. X-axis definition is missing in Fig 15.
Reviewer 2 Report
The manuscript by F. Le Guern et al. presents the preparation of the oxidised mannan and its fluorescent derivative and the binding of these species to the glycans.
The manuscript would highly benefit from an English-language edition by a native speaker or a professional editor. The current manuscript version has spelling and sentence structure issues and thus lacks readability and clarity.
As far as I am familiar with the IJMS style - the Results and Discussion sections should be separated. This separation could also improve the overall presentation of the study results.
The preparation of oxidised mannan is not fully clear: it is stated that within two hours, a 90% yield of oxidised mannan is obtained; however, the kinetic study results show approx. 25% oxidation during the same reaction time. Some clarification would help.
The data in figures 7 and 8 should be presented using a semi-logarithmic scale (as in figure 14) since the current presentation is not readily visually interpretable.
The legends in the A and B panels of figures 7 and 8 are missing.
What model was used to determine avidity? Did the authors define the avidity constants or perform other quantification to compare glycan binding to different analysed species? Could these values be compared in a summarising table?
What units are on both axes in figure 11, and on the x-axis of figure 12?
Spelling in figure 14 panel B x-axis.
Round 2
Reviewer 2 Report
In my view, the reviewed version of the manuscript by F. Le Guern et al. is significantly improved compared to the original. The authors addressed all my comments. I appreciate the updated dosing graphs (figure7 and figure8) - now the sigmoidicity of curves is visible, and the affinity/avidity can be easily compared within the experiments. The authors also separated and updated the discussion part, which made the manuscript clearer.